# Anxiety and Depression in Adolescents with Severe Asthma and in Their Parents: Preliminary Results after 1 Year of Treatment

**DOI:** 10.3390/bs9070078

**Published:** 2019-07-13

**Authors:** Amelia Licari, Riccardo Ciprandi, Gianluigi Marseglia, Giorgio Ciprandi

**Affiliations:** 1Department of Pediatrics, Fondazione IRCCS Policlinico San Matteo, University of Pavia, 27100 Pavia, Italy; 2Cystic Fibrosis Unit, Istituto G. Gaslini, 16147 Genoa, Italy; 3Allergy Clinic, Casa di Cura Villa Montallegro, 16145 Genoa, Italy

**Keywords:** severe asthma, adolescents, anxiety, depression, parents, treatment

## Abstract

Emotional problems, such as anxiety and depression, are a relevant co-morbidity in severe asthma. Anxiety and depression may also be common in the parents of asthmatic adolescents. The current study evaluated anxious and depressive symptoms in 40 adolescents suffering from severe asthma, and in their parents, before and after 1 year of treatment, tailored according to validated asthma guidelines. We used the HADS (Hospital Anxiety Depression Scale) questionnaire for the adolescents, and HADS, STAY (State-Trait Anxiety Inventory), and BDI (Beck Depression Inventory) questionnaires for their parents. We also considered the grade of asthma severity before and after 1 year of treatment. The current study demonstrated that anxiety and depression are common in both the adolescents suffering from severe asthma and their parents. Anxious and depressive symptoms were correlated between adolescents and their parents. Asthma treatment improved the asthma severity in almost all adolescents. However, the parental anxiety and depression remained unchanged at the end of the asthma treatment. Thus, a psychological assessment could be included in the adolescent severe asthma work-up, involving both the adolescents and their parents.

## 1. Introduction

Anxiety and depression may frequently affect adolescents suffering from asthma [1]. However, anxiety and depression may also be common in their parents [2]. In particular, a recent systematic review and meta-analysis considered 25 studies, including 4300 caregivers of children with asthma and 25,064 caregivers of healthy children; the conclusions pointed out that caregivers of children with asthma have greater anxious and depressive symptoms than caregivers of healthy children [3]. A vicious circle is established between the emotional disorders of the parents and anxiety and/or depression in their asthmatic children, which could, in turn, affect the asthma outcome [4]. In fact, parental anxiety and depression, mainly maternal, may negatively impact filial asthma, mainly concerning the asthma severity, asthma control, and the use of medications [4,5,6].

On the other hand, adolescent asthma may be considered a peculiar asthma phenotype that deserves adequate work-up and careful management [7,8]. Adolescence is a critical age from an emotional point of view, as adolescents are maturing in their identity and personality. Therefore, it is evident that this set of problems may also have clinical reverberations on asthma [9]. In fact, asthmatic adolescents have peculiar clinical problems concerning acceptance of the asthma diagnosis, the perception of symptoms, compliance and the adherence to the prescribed treatment, and the self-management of asthma, mainly concerning the decision to take reliever drugs [9,10].

Severe asthma is a demanding problem in clinical practice, as it is associated with poor prognosis and requires aggressive treatment [11,12]. In this regard, emotional disorders represent a relevant co-morbidity in patients with severe asthma that may negatively affect treatment outcomes [13]. Previously, a real-life study in adult asthmatic outpatients demonstrated that emotional disorders were associated with uncontrolled asthma and lower Asthma Control Test scores [14]. Therefore, we tested the hypothesis that emotional issues may be correlated with adolescents suffering from severe asthma and their parents, and that asthma treatment could affect them. On the basis of this background, the current longitudinal study aimed to investigate the effect of asthma treatment on emotional aspects, such as anxiety and depression, in the parents of adolescents with severe asthma. The study was conducted in a real-life setting, namely a tertiary level asthma clinic.

## 2. Results

This longitudinal study included 40 consecutive adolescents (22 males and 18 females; mean age 14.18 ± 1.97 years) suffering from severe asthma and their parents, evaluated before and after a 12 month treatment.

All adolescents completed the HADS questionnaire; 40 mothers and 39 fathers completed the three questionnaires.

At baseline, 23 (57.5%) asthmatic adolescents with severe asthma had anxious symptoms, as reported in Table 1. Notably, depressive symptoms were present only in patients with anxious symptoms (*p* = 0.0051). The subjects, both adolescents and their parents, were then stratified into two groups on the basis of the presence or absence of anxious symptoms in the adolescents.

The mothers of anxious adolescents were more frequently anxious, and with more severe anxious symptoms, than the mothers of non-anxious adolescents at baseline, using the three questionnaires (*p* = 0.0003, 0.0034, and 0.039, respectively), as shown in Table 1. The mothers of anxious adolescents consistently had more severe depressive symptoms than the mothers of non-anxious adolescents, using the HADS questionnaire (*p* = 0.0034).

The fathers of anxious adolescents also had more severe anxiety symptoms than the fathers of non-anxious adolescents, using the HADS questionnaire (*p* = 0.046), as shown in Table 1. The fathers of anxious adolescents consistently had more severe depressive symptoms than the fathers of non-anxious adolescents (*p* = 0.0051 and 0.024, respectively).

There were significant correlations between the anxiety of the adolescents and their parents, as well as the depression of the adolescents and their parents (Figure 1). In particular, there was a moderate (r = 0.54) relationship between anxious symptoms in the adolescents and their mothers (Figure 1A), as well as a moderate (r = 0.41) relationship between depressive symptoms in the adolescents and their mothers (Figure 1C). There was also a moderate (r = 0.52) relationship between anxious symptoms in adolescents and their fathers (Figure 1B).

After 1 year of treatment, all anxious adolescents but two had an improvement of asthma severity: 2 had mild asthma and 19 moderate asthma (Table 2).

We then analyzed the parents of anxious adolescents, comparing baseline results with follow-up results (Table 2).

The mothers of anxious asthmatic adolescents had no significant change in either anxious or depressive symptoms (*p* = ns for all items).

The fathers of anxious asthmatic adolescents consistently had no significant change in either anxious or depressive symptoms (*p* = ns for all items).

## 3. Discussion

The present study reported the clinical relevance of emotional disorders in both adolescents with severe asthma and in their parents. An emotional impairment, mainly anxiety, was common in adolescents with severe asthma. There was also a significant association between anxious and depressive symptoms: namely, depressive symptoms were present only in anxious adolescents. An emotional impairment was consistently common in the parents, mainly in mothers, with a significant association with the anxious symptoms of the adolescents. Tailored asthma treatment reduced asthma severity grade in almost all adolescents, but did not change emotional disorders in their parents.

These outcomes are consistent with the literature evidence about the frequent association between asthma and anxiety–depression [1,2,3], and confirm the important association with asthma outcomes [4,9]. In particular, Delmas evaluated 700 asthmatic teenagers, and found that asthma was associated with a higher prevalence of major depressive episodes that were in turn associated with poorer asthma control [15]. Depressive symptoms in caregivers were also associated with a higher number of primary care visits, emergency department visits, and hospital admissions in their asthmatic children [16,17]. A very recent cross-sectional study confirmed that the mothers’ depression negatively affected the lives of their asthmatic children, and was correlated with an increased number of emergency department visits [18]. On the other hand, it has been reported that the psychological health of the parents is strongly affected by their child’s chronic disease [19,20]. Moreover, Wamboldt evaluated the parents, usually the mothers, of 62 adolescents admitted to a tertiary care asthma center for severe asthma, and showed a link between severe asthma and familial affective disorders [21]. Akcakaya investigated the relationship between the severity and duration of asthma and psychological problems in 57 asthmatic children, as well as the probability of maternal anxiety [22]. Emotional factors and family dynamics were found to be triggering factors for asthma attacks, and were positively correlated with asthma severity. Both asthmatic children and their mothers were negatively affected by the disease. Ortega studied the associations between parental mental health problems and asthma attacks in a group of Puerto Rican youths [23]. Parents with mental health problems were more likely to report histories of asthma attacks in their children compared with parents without mental health problems. Kean demonstrated that 49 adolescents who had experienced a life-threatening asthma episode and their parents both had high levels of post-traumatic stress symptoms that were linked to asthma morbidity [24]. Yuksel reported that the anxiety and depression symptoms of the mothers of 75 asthmatic youths were significantly more severe than in the mothers of healthy subjects [25]. Rockhill demonstrated that asthmatic adolescents without behavioral problems and with less severe anxiety and depression were recognized significantly less often by their parents [26]. Szabo showed that caregivers of asthmatic youths have more depressive, and consequently anxious symptoms than the average Hungarian population [27]. Guxen reported that maternal psychological distress during pregnancy was associated with increased odds of wheezing in their children during the first 6 years of life, independent of paternal psychological distress during pregnancy and maternal and paternal psychological distress after delivery [28]. Lau reported that poorly controlled asthma in adolescents was associated with maternal anxiety [29]. Interestingly, maternal anxiety may induce negative behavioral effects: Dantas reported that a high proportion of the mothers of asthmatic adolescents restrained their children from engaging in physical activity [30].

Therefore, there is convincing evidence that emotional disorders are important in both asthmatic adolescents and their parents. In this regard, the current study provided findings confirming the literature data.

Interestingly, the asthma treatment did not change anxious and depressive symptoms in the parents, despite the positive effect on adolescent asthma. This finding has two main possible explanations: that the emotional problems are scarcely modifiable, and the parental awareness that asthma is a chronic disorder that is lifelong even though manageable.

However, the present study has two principal limitations, including the lack of a control group and the administration of psychological questionnaires only, without a thorough psychological assessment.

On the contrary, three questionnaires were together evaluated, adolescents and parents were also considered simultaneously, and the data were collected in a real-life setting, so the findings may mirror what occurs in daily practice.

In fact, these outcomes suggest that a psychological assessment should be recommendable in all adolescents suffering from asthma and in their parents. In addition, a psychological treatment, such as psychotherapy, could be helpful to improve anxious and depressive symptoms in both asthmatic adolescents and their parents.

The possible future perspective could be the inclusion of psychological intervention in an algorithmic approach for the treatment of severe asthma, as recently provided [31]. In addition, psychological support may be fruitful in communication strategies to improve the doctor–patient relationship, as recently suggested by a consensus in pediatric severe asthma [32].

## 4. Materials and Methods

This longitudinal real-life study enrolled a series of consecutive adolescents with asthma visiting for the first time at the third-level pediatric clinic of the Policlinico San Matteo of Pavia (Italy).

Inclusion criteria were: age between 12 and 17 years, both genders, and severe asthma grade. Exclusion criteria were: use of concomitant medications and severe chronic disorders able to interfere with the interpretation of the results.

The procedure was approved by the Ethics Committee of the Istituto Giannina Gaslini of Genoa (code number: 22253/2017; in the context of the Italian Project “ControL’Asma” promoted by the Italian Society of Paediatric Allergy and Immunology). Both the parents signed an informed consent, routinely administered to all outpatients (the parents for minors) visiting the Policlinico San Matteo of Pavia.

Asthma diagnosis was confirmed according to the 2018 Global Initiative for Asthma (GINA) document [11]. The asthma severity was measured according to the International Guidelines on Severe Asthma [12].

Adolescents and their parents were evaluated before and after a 12 month treatment. Adolescents’ treatment was tailored according to the Global Initiative for Asthma (GINA) guidelines [11].

Patients and parents were re-evaluated after 12 months.

Anxiety and depression aspects were evaluated in adolescents with asthma and in their parents.

The adolescents completed the Hospital Anxiety Depression Scale (HADS) questionnaire alone, and the parents completed three psychometric questionnaires: HADS, BDI, and STAI. The questionnaires were administered during the visits.

The Hospital Anxiety Depression Scale (HADS) gives clinically meaningful results as a psychological screening in clinical group comparisons [33]. In interpretation of the questionnaire, a score of >7 (in the two subscales) has been found to define anxious or depressive symptoms [34].

The Beck Depression Inventory II (BDI-II) is a validated, 21 item, self-administered questionnaire to measure depression [35]. Each question has four choices, ranging in point value from 0 to 3. Total scores of 0 to 13 represent no depression; 14 to 19, mild depression; 20 to 28, moderate depression; and 29 to 63, severe depression.

The State-Trait Anxiety Inventory (STAI) questionnaire measures both the present (state: STAI-Y1) and the trait (trait: STAI-Y2) feelings of some characteristics of anxiety, including apprehension, tension, nervousness, and worry [36]. The 40 item STAI-Y scores range from 20 to 80. Weighted scores for 20 items on each scale are added together to give total anxiety scores ranging from 20–80 (most anxious), and scores higher than 65 indicate clinically relevant anxiety.

### Statistical Analysis

Data were reported as median with inter-quartile range or as absolute and relative (percentages) numbers. The Wilcoxon signed rank test and Spearman’s test were used. Statistica software 9.0 (StatSoft Corp., Tulsa, OK, USA) was used.

## 5. Conclusions

The current study demonstrated that anxiety and depression are common in adolescents suffering from severe asthma and in their parents, mainly in mothers. Thus, psychological intervention could be helpfully included in the adolescent severe asthma work-up, considering the entire family.

## Figures and Tables

**Figure 1 behavsci-09-00078-f001:**
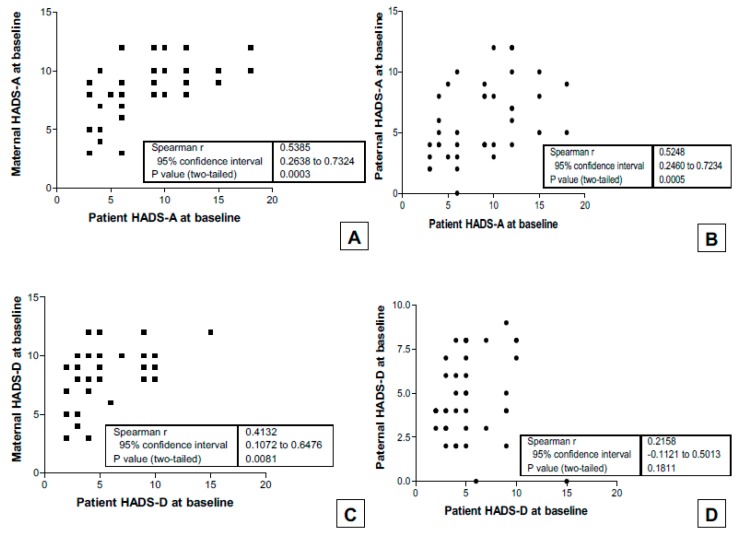
(**A**) Relationship between HADS-A in the mothers and in the adolescents at baseline; (**B**) relationship between HADS-A in the fathers and in the adolescents at baseline; (**C**) relationship between HADS-D in the mothers and in the adolescents at baseline; (**D**) the relationship between HADS-D in the fathers and in the adolescents at baseline.

**Table 1 behavsci-09-00078-t001:** Emotional data in non-anxious and anxious adolescents with severe asthma and their parents at baseline.

At Baseline		Non-Anxious	Anxious	*p* Value
**Adolescents**		**17**	**23**	
Depression (N. %)		(0%)	9 (39.13%)	**0.0051**
**Mothers**				
Anxiety HADS-A (N. %)		9 (52.94%)	23 (100%)	**0.0003**
Anxiety STAI-Y1 (N. %)	Absent	9 (52.94%)	1 (4.35%)	**0.0034**
	Mild	2 (11.76%)	4 (17.39%)	
	Moderate	6 (35.29%)	14 (60.87%)	
	Severe	(0%)	4 (17.39%)	
Anxiety STAI-Y2 (N. %)	Absent	9 (52.94%)	3 (13.04%)	**0.039**
	Mild	2 (11.76%)	3 (13.04%)	
	Moderate	6 (35.29%)	15 (65.22%)	
	Severe	(0%)	2 (8.7%)	
Depression HADS-D (N. %)		6 (35.29%)	19 (82.61%)	**0.0034**
Depression BDI-II (N. %)	Absent	9 (52.94%)	4 (17.39%)	**0.0597**
	Mild	6 (35.29%)	14 (60.87%)	
	Mild-to-moderate	2 (11.76%)	5 (21.74%)	
**Fathers**				
Anxiety HADS-A (N. %)		3 (17.65%)	12 (52.17%)	**0.0464**
Anxiety STAI-Y1 (N. %)	Absent	15 (88.24%)	12 (54.55%)	0.068
	Moderate	2 (11.76%)	8 (36.36%)	
	Severe	(0%)	2 (9.09%)	
Anxiety STAI-Y2 (N. %)	Absent	15 (88,24%)	13 (59.09%)	0.081
	Moderate	1 (588%)	8 (36.36%)	
	Severe	1 (5.88%)	1 (4.55%)	
Depression HADS-D (N. %)		(0%)	9 (39.13%)	**0.0051**
Depression BDI-II (N. %)	Absent	16 (94.12%)	13 (59.09%)	**0.024**
	Mild	1 (5.88%)	9 (40.91%)	

**Table 2 behavsci-09-00078-t002:** Emotional data of the parents of anxious adolescents with severe asthma at baseline and at 12 month follow-up (FU).

Adolescents		At Baseline	At 12 mo FU	*p*-Value
Asthma severity				
Mild		0	2 (8.7%)	
Moderate		0	19 (82.61%)	
Severe		23 (100%)	2 (8.7%)	
**Mothers**				
Anxiety HADS-A (N. %)		23 (100%)	21 (91.3%)	0.49
Anxiety STAI-Y1 (N. %)	Absent	1 (4.35%)	1 (4.35%)	-
	Mild	4 (17.39%)	4 (17.39%)	
	Moderate	14 (60.87%)	14 (60.87%)	
	Severe	4 (17.39%)	4 (17.39%)	
Anxiety STAI-Y2 (N. %)	Absent	3 (13.04%)	3 (13.04%)	-
	Mild	3 (13.04%)	3 (13.04%)	
	Moderate	15 (65.22%)	15 (65.22%)	
	Severe	2 (8.70%)	2 (8.70%)	
Depression HADS-D (N. %)		19 (82.61%)	17 (73.91%)	0.72
Depression BDI-II (N. %)	Absent	4 (17.39%)	4 (17.39%)	-
	Mild	14 (60.87%)	14 (60.87%)	
	Mild-to-moderate	5 (21.74%)	5 (21.74%)	
**Fathers**				
Anxiety HADS-A (N. %)		12 (52.17%)	11 (47.83%)	-
Anxiety STAI-Y1 (N. %)	Absent	12 (52.17%)	12 (52.17%)	-
	Mild	0	0	
	Moderate	8 (34.78%)	8 (34.78%)	
	Severe	2 (8.70%)	2 (8.70%)	
Anxiety STAI-Y2 (N. %)	Absent	13 (56.52%)	13 (56.52%)	-
	Mild	8 (34.78%)	8 (34.78%)	
	Moderate	0	0	
	Severe	1 (4.35%)	1 (4.35%)	
Depression HADS-D (N. %)		9 (39.13%)	8 (34.78%)	
Depression BDI-II (N. %)	Absent	12 (52.17%)	12 (52.17%)	-
	Mild	0	0	
	Mild-to-moderate	8 (34.78%)	8 (34.78%)	
	Severe	2 (8.69%)	2 (8.69%)

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
