# Peer review of "Anxiety and Depression in Adolescents with Severe Asthma and in Their Parents: Preliminary Results after 1 Year of Treatment"

_behavsci, 2019, doi:10.3390/bs9070078_

Round 1
Reviewer 1 Report
The present manuscript is a brief report on a longitudinal study focused on emotional aspects (anxiety and depression) in the parents of adolescent asthma patients. As this manuscript is submitted as a “brief report” and not a full research manuscript, one could not expect a very detailed derivation of the research from the current state of the art in this field. The same applies to the discussion. However, what I do miss is a compact short paragraph in the introduction that cites the latest related studies. These studies should also be taken up in the discussion. In the revision, the authors should address these relevant sources a bit more explicitly. In terms of methods, I do not see any severe points of criticism.
Author Response
We implemented the Introduction and Discussion citing and discussing more studies.
Reviewer 2 Report
The paper “Anxiety and depression in parents of adolescents with severe asthma” by Licari et al. would have the purpose to evaluate anxiety and depression in parents of 40 adolescents suffering from severe asthma. The subject of the paper could be interesting, but problems in conceptualization and methodology are relevant. Thus, some critical points have to be taken into consideration:
- The paper does not seem in line with manuscript preparation instruction for authors. I suggest the authors to revise the structure of the paper;
- The background on which the study conceptualization is based seems extremely poor, both for psychopathological context (i.e. authors describe anxiety and depression in adolescents and their parents without differentiate these disorders in the two samples, and it is not clear if they mean specific clinical pictures or symptoms) and for relationship between anxiety and depression in parents and anxiety and depression in their asthmatic children. I suggest authors to structure the introduction section referring to current professional literature;
- The sentences at lines 37-40 should be reported in the results paragraphs as socio-demographic characteristics of the sample;
- What does “GINA” mean? Authors should report the meaning of the acronym and mention some references on these guidelines;
- Moreover, the design and the type of the study should be clarified. It is also evident a lack of clarity in inclusion and exclusion criteria for patients, which they have to be better specified;
- Since the authors declare that the study has been approved by the Ethics Committee, I suggest to report the number and the reference of protocol. It could be useful also to provide a copy of the informed consent to trial participation for parents;
- Literature references on rating scales used are totally missing. I suggest authors to report them in methods section;
- One of the major problems of the paper, in my opinion, are scopes and endpoints of the study. Authors aimed to evaluate anxiety and depression in parents of adolescents with asthma but the paper does not mention the purpose of the research or any possible implications of this relationship; for example, the influence of anxiety and depression in parents on outcome in asthma treatment in children could be an interesting endpoint to be considered. The current scope of the study doesn’t seem clearly defined;
- The sample size is very small. I suggest the authors to report a sample size evaluation in order to consider properly the statistical power of the findings;
- The possible explanation reported by the authors on the relevance of anxiety and depression in parents of adolescent with asthma at lines 84-87 is questionable and lacking. Moreover, also the sentence at lines 88-89 should be better contextualized;
- Statement reported at lines 94-97 does not take into account any possible future perspective;
- The literature references list should be expanded, referring to the current literature on the topic of the article;
In light of these considerations, the study should be redesigned in methodology and the paper should be entirely re-written in order to achieve a version suitable for publication.

Author Response
- The paper does not seem in line with manuscript preparation instruction for authors. I suggest the authors to revise the structure of the paper;
R We revised the structure of the paper.
- The background on which the study conceptualization is based seems extremely poor, both for psychopathological context (i.e. authors describe anxiety and depression in adolescents and their parents without differentiate these disorders in the two samples, and it is not clear if they mean specific clinical pictures or symptoms) and for relationship between anxiety and depression in parents and anxiety and depression in their asthmatic children. I suggest authors to structure the introduction section referring to current professional literature;
R We revised the text according with these suggestions.
- The sentences at lines 37-40 should be reported in the results paragraphs as socio-demographic characteristics of the sample;
R We moved the sentence as suggested.
- What does “GINA” mean? Authors should report the meaning of the acronym and mention some references on these guidelines;
R We provided the meaning, it was cited as the reference N° 8.
- Moreover, the design and the type of the study should be clarified. It is also evident a lack of clarity in inclusion and exclusion criteria for patients, which they have to be better specified;
R We provided this information
- Since the authors declare that the study has been approved by the Ethics Committee, I suggest to report the number and the reference of protocol. It could be useful also to provide a copy of the informed consent to trial participation for parents;
R We provided the number of the protocol. It concerned a Research Project involving several Italian center that are investigating asthma in the pediatric age.
- Literature references on rating scales used are totally missing. I suggest authors to report them in methods section;
R We reported the references
- One of the major problems of the paper, in my opinion, are scopes and endpoints of the study. Authors aimed to evaluate anxiety and depression in parents of adolescents with asthma but the paper does not mention the purpose of the research or any possible implications of this relationship; for example, the influence of anxiety and depression in parents on outcome in asthma treatment in children could be an interesting endpoint to be considered. The current scope of the study doesn’t seem clearly defined;
R We revised the manuscript taking into account these points providing more information and details.
- The sample size is very small. I suggest the authors to report a sample size evaluation in order to consider properly the statistical power of the findings;
R Actually, we investigated adolescents with severe asthma. As the percentage of children with severe asthma is <5%, we have had to screen about 7-800 adolescents with asthma in a time frame lasting >1 year. Anyway, the current paper is a preliminary report.
- The possible explanation reported by the authors on the relevance of anxiety and depression in parents of adolescent with asthma at lines 84-87 is questionable and lacking.
R We expanded this topic.
- Moreover, also the sentence at lines 88-89 should be better contextualized;
R We revised this sentence according with the comment.
- Statement reported at lines 94-97 does not take into account any possible future perspective;
R We added this issue as requested.
- The literature references list should be expanded, referring to the current literature on the topic of the article;
R We provided more references
In light of these considerations, the study should be redesigned in methodology and the paper should be entirely re-written in order to achieve a version suitable for publication.
R We revised the manuscript considering these suggestions.
Round 2
Reviewer 1 Report
In the revised version of the manuscript, the authors addressed the major points. It would have been nice to read a response letter where the authors comment in detail how they reacted to to the points, instead of just answering in one single sentence. However, the authors corrected the major points in the revised version and, so, it would be unfair from my side not to recommend the new version for publication
Reviewer 2 Report
None